# Effect of Rice Grain (*Oryza sativa* L.) Enrichment with Selenium on Foliar Leaf Gas Exchanges and Accumulation of Nutrients

**DOI:** 10.3390/plants10020288

**Published:** 2021-02-03

**Authors:** Ana Coelho Marques, Fernando C. Lidon, Ana Rita F. Coelho, Cláudia Campos Pessoa, Inês Carmo Luís, Paula Scotti Campos, Manuela Simões, Ana Sofia Almeida, Maria F. Pessoa, Carlos Galhano, Mauro Guerra, Roberta G. Leitão, Paulo Legoinha, José C. Ramalho, José N. Semedo, Ana Paula Rodrigues, Paula Marques, Cátia Silva, Ana Ribeiro-Barros, Maria José Silva, Maria Manuela Silva, Karliana Oliveira, David Ferreira, Isabel P. Pais, Fernando H. Reboredo

**Affiliations:** 1Earth Sciences Department, Faculdade de Ciências e Tecnologia, Universidade Nova de Lisboa, Campus da Caparica, 2829-516 Caparica, Portugal; fjl@fct.unl.pt (F.C.L.); arf.coelho@campus.fct.unl.pt (A.R.F.C.); c.pessoa@campus.fct.unl.pt (C.C.P.); idc.rodrigues@campus.fct.unl.pt (I.C.L.); mmsr@fct.unl.pt (M.S.); mfgp@fct.unl.pt (M.F.P.); acag@fct.unl.pt (C.G.); pal@fct.unl.pt (P.L.); djo.ferreira@campus.fct.unl.pt (D.F.); fhr@fct.unl.pt (F.H.R.); 2GeoBioTec Research Center, Faculdade de Ciências e Tecnologia, Universidade Nova de Lisboa, Campus da Caparica, 2829-516 Caparica, Portugal; paula.scotti@iniav.pt (P.S.C.); sofia.almeida@iniav.pt (A.S.A.); cochichor@mail.telepac.pt (J.C.R.); jose.semedo@iniav.pt (J.N.S.); aribeiro@isa.ulisboa.pt (A.R.-B.); mjsilva@isa.ulisboa.pt (M.J.S.); karliana.oliveira@ipbeja.pt (K.O.); isabel.pais@iniav.pt (I.P.P.); 3Instituto Nacional de Investigação Agrária e Veterinária, I.P. (INIAV), Avenida da República, Quinta do Marquês, 2780-157 Oeiras, Portugal; 4Instituto Nacional de Investigação Agrária e Veterinária, I.P. (INIAV), Estrada de Gil Vaz 6, 7351-901 Elvas, Portugal; 5LIBPhys, Physics Department, Faculdade de Ciências e Tecnologia, Universidade Nova de Lisboa, Campus da Caparica, 2829-516 Caparica, Portugal; mguerra@fct.unl.pt (M.G.); rg.leitao@fct.unl.pt (R.G.L.); 6PlantStress & Biodiversity Lab., Centro de Estudos Florestais (CEF), Instituto Superior Agronomia (ISA), Universidade de Lisboa (ULisboa), Quinta do Marquês, Av. República, 2784-505, Oeiras and Tapada da Ajuda, 1349-017 Lisboa, Portugal; anadr@isa.ulisboa.pt; 7Centro Operativo e Tecnológico do Arroz (COTARROZ), 2120-014 Salvaterra de Magos, Portugal; p.marques@cotarroz.pt (P.M.); catia.leonardo.silva@gmail.com (C.S.); 8Escola Superior de Educação Almeida Garrett (ESEAG-COFAC), Avenida do Campo Grande 376, 1749-024 Lisboa, Portugal; abreusilva.manuela@gmail.com; 9Instituto Politécnico de Beja (IPBeja), 7800-295 Beja, Portugal

**Keywords:** elemental composition, photosynthesis, rice cultivars, selenium biofortification

## Abstract

An agronomic itinerary for Se biofortification of two rice cultivars (Ariete and Ceres) through foliar fertilization with sodium selenate and sodium selenite with different concentrations (25, 50, 75 and 100 g Se.ha^−1^), was implemented in experimental fields. The selenium toxicity threshold was not exceeded, as shown by the eco-physiological data obtained through leaf gas exchanges. The highest Se enrichment in paddy grains was obtained with selenite for both cultivars, especially at the highest doses, i.e., 75 and 100 g Se.ha^−1^, with approximately a 5.0-fold increase compared with control values. In paddy grains, Zn was the most affected element by the treatments with Se with decreases up to 54%. When comparing the losses between rough and polished grains regardless of the cultivars, Se species and concentrations, it was observed that only Cu, Mg and Zn exhibited losses <50%. The remaining elements generally had losses >70%. The loss of Se is more pronounced in Ceres cultivar than in Ariete but rarely exceeds 50%. The analysis by µ-EDXRF showed that, in Ariete cultivar, Se is mostly homogeneously distributed in the grain regardless of any treatments, while in Ceres cultivar, the Se distribution seems to favor accumulation in the periphery, perhaps in the bran.

## 1. Introduction

Selenium is an essential element in the human diet, although its presence in plants is scarce [1]. The application of Se in the form of sodium selenate [2,3] or sodium selenite [4] is a useful and well-known method to increase the Se concentration in food crops. Studies show that foliar spraying with Se is more viable and effective than its application to the soil [5]. Roots can mobilize Se in the forms of selenate, selenite, elemental Se and organic compounds such as SeCys (seleno-cysteine) and SeMeth (seleno-methionine) [6], although soil speciation is regulated by both pH and Eh—selenite is the main form in anaerobic soils with a neutral to acidic pH, as occurs in paddy fields, while selenate is in the main water soluble form in oxic soils [6].

Foliar spraying is affected by leaf anatomy beyond substrata characteristics and environmental factors. In general, after foliar Fe application, it takes between 10 and 20 days to absorb 50% of the micronutrient [7], although for Ca, Mn and Zn only 1–2 days were needed for 50% absorption, which agrees with the promising results of foliar fertilizer application to improve grain Zn in wheat and other cereals [8]. The effectiveness of foliar feeding was seen after evaluation of the Se rice grain concentrations which increased to 0.471–0.640 μg g^−1^, after using a Se enriched fertilizer at a rate of 20 g of Se.ha^−1^, in the forms of sodium selenite and sodium selenate, when the average Se content of regular polished rice in China is 0.025 ± 0.011 μg g^−1^ [9].

Several studies confirmed that selenate is more effective than selenite (applied either through soil or foliar spray) regarding cereal enrichment [10,11,12] while others point out to selenite [13,14]. Thus, the forms of Se supply but also genotype differences, plus the substrata characteristics affect the Se bioavailability by plant cultivars.

The Se of selenate is more easily transported to the aerial part while the Se of selenite tends to accumulate in the roots, indicating different absorption and mobility routes [15]. Hawrylak-Nowak observed that the Se concentration in lettuce (edible part) increased as a function of the increasing concentrations of selenite and selenate applied in hydroponic cultures, with values ranging between 3.7 and 30.6 and 4.7 and 43.3 mg kg^−1^, for selenite and selenate, respectively [16]. Conversely, the concentrations of Se in the roots ranged between 48.5 and 201.4 and 1.9 and 44.2 mg kg^−1^, for selenite and selenate, respectively, clearly indicating that the translocation to the aerial organs is poor when selenite is used, but not the uptake.

Thus, through agronomic biofortification it is possible to increase the intake of Se by humans, due to the Se enrichment of main food crops such as rice [14,17,18,19] wheat [2,20] potato [21] and lettuce [16], for example. Selenium is also beneficial in defending plants against oxidative stress caused by oxygen radicals produced by internal metabolic or external factors [22] also delaying the effects of senescence at appropriate levels.

Regarding the interaction of Se with other nutrients, studies with biofortified wheat plants with different concentrations of selenate and selenite have shown that micronutrients were more affected by the presence of Se than macronutrients, and the elements that accumulate more in the roots (Ca, Fe, Mn and Zn) were more sensitive to the presence of Se [23]. The rice plant has a tendency to absorb Fe because ferrous ion is abundant in flooded soils and reduced iron is easily absorbed [24]. Moreover, the assimilation of Se into organoselenium compounds is supposed to compete with the assimilation of S [25], while the application of P fertilizer to the soil increases the concentration of Se in wheat in selenate-fertilized calcareous soils, but not in selenite-fertilized [26].

The accumulation of Mn, Fe and Zn in two wheat seedling cultivars showed that the plants supplied with 2 µM Na_2_SeO_4_ over two weeks did not exhibit significant differences in the overwhelming majority of the cases, regarding the Mn, Fe and Zn content, compared with the control plants [27].

The japonica rice Ariete is appreciated by farmers because of its production and quality. Ariete has been widely cultivated throughout Europe, and today it ranks as the main cultivar in Portugal, accounting for 40% of the total area cultivated in 2010 [28]. Each year, tests are carried out on new genetic improvement cultivars with a view to introducing and replacing cultivars that meet the needs of producers and industry. Ceres was the first Portuguese cultivar of carolino rice to be listed in the national catalog of cultivars. Ariete and Ceres rice grains are classified as long (grain length/width ratio of 2.6 and 2.5), with average amylose of 15.3% and 18.2%, respectively [29]. Considering the importance of *Oryza sativa* L. worldwide, this study aimed to develop an agronomic itinerary leading to Se rice enrichment, while evaluating photosynthetic metabolism, and the impact of the application of Se on macro and micronutrients rice grains, in both paddy, brown and polished forms. Furthermore, the distribution of Se in grain was also studied by µ-EDXRF.

## 2. Results

### 2.1. Physiological Monitoring during Biofortification

Physiological data were acquired after the second foliar Se fertilization in rice leaves treated with the two highest concentrations of selenite and selenate, beyond the controls.

The Ariete plants did not show any significant impact on net photosynthesis (P_n_), regardless of the form (Na_2_SeO_4_ or Na_2_SeO_3_) or dose (50 or 100 g Se.ha^−1^), despite a minor tendency of higher values in the Se-Treated plants (Figure 1). In contrast, all Se-treated plants showed significant increases in stomatal conductance to water vapor (g_s_), in particular in those of the 100 g Se.ha^−1^ of Na_2_SeO_3_ treatment which doubled its value. This g_s_ rise was paralleled with significant transpiration (E) increases, whereas instantaneous water use efficiency (iWUE) was reduced significantly for both treatments and cultivars.

The cultivar Ceres showed a similar pattern of changes to that of Ariete as regards the leaf gas exchanges. After two foliar applications, there was no negative impact on P_n_ of any of the Se-treatments. In fact, P_n_ was significantly increased by the 50 g Se.ha^−1^ of Na_2_SeO_4_ treatment, and, especially with the maximum doses of both selenite and selenite compounds, always as compared to the control (Figure 1). Plants treated with the highest selenite or selenate concentrations showed a tendency to display higher g_s_ and E values, although significantly only for Na_2_SeO_3_ cultivar. Still, no significant impacts were found for iWUE in Ceres (Figure 1).

### 2.2. Accumulation of Selenium in Rice Grain

It was found that, at harvest, the average yields (kg ha^−1^) were, for Ariete, 4810 and 4548 and, for Ceres, 5499 and 5322 (for both cultivars, after application of selenate and selenite, respectively). In general, an increase in Se concentration in the paddy grain of both cultivars as a function of increasing doses of foliar spraying was observed, although the highest concentrations were noticed for selenite (Na_2_SeO_3_) at a dose of 100 g Se.ha^−1^. It is interesting to note a growing enrichment of Se grains as the concentration of Se-fertilizer increases, although in Ceres cultivar treated with selenite it seems that a threshold enrichment was reached since the concentrations verified with 50, 75 and 100 g Se.ha^−1^ are very similar (Table 1).

If we take into account the higher treatments (75 g Se.ha^−1^, 100 g Se.ha^−1^), in all cases, the concentrations of Se in the grains of both cultivars treated with selenite (Na_2_SeO_3_) contained more Se than those treated with selenate (Na_2_SeO_4_). The levels of Se in the control grains were very low generally around 2 mg kg^−1^.

The concentrations of Se within grain cultivars often decrease as the grains are dehusked and polished. White grains (polished) from Ceres cultivar treated with 75 g and 100 g Se.ha^−1^ selenate contained 43.8% and 54.4% less Se than equivalent treatments in brown grains, and 56.6% and 62.9% less if paddy grains were considered. In the same context (cultivar and concentrations) the decrease in Se in polished grains when selenite was applied was a little bit lower, i.e., 8% and 13.4% in 75 g and 100 g Se.ha^−1^, respectively (Table 1, Table 2 and Table 3).

Regarding the Ariete cultivar we did not find, in general, a pronounced decrease in Se concentrations throughout the processing phase, in selenate treated plants, while the loss in selenite treated plants range between 29% and 32.4% (Table 3). In two particular cases a small Se enrichment was observed in polished grains—paddy grains derived from plants treated with 100 g Se.ha^−1^ selenate and 75 g Se.ha^−1^ selenite, which contained 8.10 and 9.88 mg kg^−1^ Se, respectively, whereas the polished grains with the same treatments contained 8.22 and 10.4 mg kg^−1^ Se, respectively, with the correspondent enrichment of 1.5% and 5.3% (Table 1 and Table 3), enhancing the great variability of Se concentrations. A similar situation occurs in control plants in polished grains of both cultivars, with enrichments of Se around 20%. This fact is mainly attributed to the very low concentrations of Se (circa. 2 mg.kg^−1^) whose variability induces those increments.

### 2.3. Macro and Micronutrient Quantification in Rice Grains

#### 2.3.1. Paddy Rice

Regarding the accumulation of Cu as a function of the different concentrations of Se, as selenite and selenate, no major variations were detected as a result of the different treatments, regardless of the cultivars. The average values were not significantly different at the 0.05 significance level. Nevertheless, Ceres cultivar accumulated more Cu than Ariete cultivar. The minimum and maximum concentrations range between 4.85 and 5.94 in Ariete while in Ceres they were between 5.75 and 7.74 mg kg^−1^ (Table 1).

Regarding Zn and Fe, it was noted that in Ariete, the Fe levels were similar, ranging between 31.0 and 39.6 mg kg^−1^, when considering the whole treatments and different types of application, while in Ceres cultivar these levels range between 37.8 and 51.0 mg kg^−1^, regardless of the treatment used. Zinc was the element clearly affected by the treatments with Se—control grains exhibited approximately 50 and 70 mg kg^−1^ for Ariete and Ceres, respectively. When the higher concentrations of Se were applied (75 g and 100 g Se.ha^−1^ of both selenate and selenite), the decrease in Zn in Ariete grains was ca. 36%, while in Ceres the reduction range was between 29% and 54% (Table 1).

Regarding Ca, the concentrations in the different grains are not significantly different (*p* ≤ 0.05), although the values observed in the Ariete cultivar treated with selenite are higher than those observed for selenate. Conversely, in Ceres cultivar the higher concentrations were noticed for selenate (Table 2). The concentrations of Mg in the paddy grains range between 408 and 457 mg kg^−1^ regardless of the cultivars, Se species and concentrations used, thus indicating a narrow variation. The concentrations of K in the paddy grains of Ariete cultivar range between 6300 and 9400 mg kg^−1^, while in Ceres cultivar they range between 5500 and 8700 mg kg^−1^. In general, the treatments did not seem to influence the K concentration in grain by the cultivars, although in Ariete the control grains presented generally the lowest levels (Table 1).

#### 2.3.2. Brown Rice

In general, the concentrations of Fe and Cu did not vary significantly as a result of the different treatments (selenite and selenate) and concentrations applied and regardless of the cultivars (Table 2). Regarding Zn, a strong decrease was observed in Ariete cultivar, whose treated grains contained Zn levels ranging between 24.9 and 28.9 mg kg^−1^ when selenate was applied, and 26.1 and 28.3 mg kg^−1^ when selenite was used—control grains presented 29.5 mg kg^−1^ (Table 2). In Ceres cultivar the Zn concentrations ranged between 28.2 and 39.0 mg kg^−1^ (selenate) and 28.5 and 37.6 mg kg^−1^ (selenite), whose values are closer to the control one which is 44.6 mg kg^−1^. The Ca concentrations in the grains exhibited a great variation regardless of the treatments (selenite vs. selenate) and concentrations used; thus, we cannot establish a trend regarding this macronutrient, although the mean values in Ceres were clearly higher than those observed for Ariete (Table 2).

The Mg concentrations in the grains were close regardless of the cultivars, type of Se fertilizer and concentration used. In Ariete cultivar the concentrations range between 425 and 454 mg kg^−1^ regardless of the treatments, while in Ceres the range is smaller, i.e., between 442 and 459 mg kg^−1^ (Table 2). No significant differences were observed (*p* ≤ 0.05) in the concentrations of K in both Se treated Ariete and Ceres cultivars. The control grains exhibited the lowest K levels in Ariete, but not in Ceres cultivar (Table 2).

#### 2.3.3. White Rice

In polished rice, the concentrations of Fe and Cu the grains were not significantly different at the 0.05 significance level, regardless of the cultivars, type of treatment and concentrations used (Table 3). Similarly to paddy and brown grains, the Zn levels in control grains were always higher than levels observed in the different treatments. Nevertheless, the gap was not as large as that which occurred for paddy and brown grains. For example, in Ariete cultivar grain Zn concentrations range between 18.3 and 24.4 mg kg^−1^ regardless of the treatment and concentration used, while in Ceres cultivar this range is between is 21.9 and 29.5 mg kg^−1^. Control grains of Ariete and Ceres contained an average concentration of 32.5 mg kg^−1^ and 45.8 mg kg^−1^, respectively (Table 3).

The Ca levels are low in both cultivars, seldom exceeding 20.0 mg kg^−1^. Additionally, they exhibited a great variability within treatments. The K concentrations in polished grains range between 1600 and 2200 mg kg^−1^ and 1600 and 2800 mg kg^−1^, for Ariete and Ceres cultivars, respectively, while control values are 1700 mg kg^−1^ for Ariete and 2100 mg kg^−1^ for Ceres. Despite the highest Mg levels being observed in control grains, the concentrations in general are not significantly different (*p* ≤ 0.05) (Table 3).

#### 2.3.4. Comparison between Elemental Levels in Paddy and White Rice Grains

In white rice, the milling and polishing processes remove nutrients; thus, it is interesting to compare final composition in terms of several micro and macronutrients vis a vis the composition observed in paddy rice, i.e., the whole rice grain with the hulls. This analysis is focused on the losses of both cultivars (Ariete and Ceres) regardless of treatments with selenite and selenate, or even control plants. In Ariete, the minimum losses were observed for Cu and Zn with percent values ranging between 13.6% and 27.8% for Cu and 13.8% and 42.7% for Zn. In the case of Cu, if we discount the lowest value a certain constancy was observed regardless of the cultivars with values ranging between 20% and 27.8%. In Ceres, the losses for the same elements were more pronounced ranging between 17.4% and 45.8% for Cu and 20% and 44.1% for Zn. In the case of selenate treated plants the loss of Cu varies between 31.1% and 42.1% while in selenite treated plants between 17.4% and 45.8%, thus exhibiting a large variability (Table 3). The loss of Fe in white grains was drastic with levels, in the overwhelming majority of the cases, below 10 mg kg^−1^ with correspondent losses ranging between 64.5% and 79.1% for Ariete and 75.5% and 88.1% for Ceres. Regarding Zn, a decrease in the concentration was noted when polished grains were analyzed, although this trend was not so pronounced as that which was verified for Fe (Table 3).

Regarding macronutrients, remarkable losses were noted for Ca with a narrow variability in Ariete cultivar treated with selenite (67.9–78.8%) and Ceres treated with selenate (66.5–77.8%). Conversely, selenate treated plants from Ariete and selenite treated plants from Ceres exhibited a larger variation with values ranging between 32.8% and 67.9% for Ariete and 53.9% and 79.8% for Ceres. The losses of K associated with polished grains were generally above 70%, for both cultivars, while the losses of Mg range between 25.5% and 49.5%, regardless of cultivars, Se species and concentrations used. Similarly to the other elements, losses of Se also occur, although showing a great variability—between 8.05% and 62.9% in Ceres, and 32.4% and an increment of 21.9% in Ariete (Table 3).

### 2.4. Colour Analysis

In paddy rice, L values are so close that no significant differences can be observed (*p* ≤ 0.05), a finding that is extensive in brown and white grains. Nevertheless, the L levels in Ceres cultivar are a little bit smaller than the corresponding values of Ariete.

Regarding brown rice, the value of L in Ariete and Ceres cultivars are almost identical regardless of the type of application (selenate, selenite) and concentrations used (Table 4). A similar situation occurred with polished rice although it can be stressed that L values show an increasing trend from paddy to white grains. The a* ( red—green transitions) positive values in paddy and brown rice are not significantly different (*p*≤ 0.05), regardless of the cultivars and Se concentrations, while negative values were observed for polished rice (*p* ≤ 0.05)—in Ariete cultivar these negative values range between −0.60 and −0.88, while in the Ceres cultivar the range is between −0.81 and −1.12. In paddy rice the b* (yellow—blue transitions) are not significantly different (*p* ≤ 0.05), regardless of the cultivars and different Se concentrations used, although b* values in Ceres cultivar are a little bit higher than similar values derived from Ariete (Table 4).

When brown rice is considered we saw a small decrease in b* values in both cultivars followed by a strong reduction in b* values in white rice (>50%), which is probably related with industrial processes such as dehusking and whitening (Table 4).

### 2.5. Location of Se in Grains

The location of Se in rice grains was performed on dehusked grains since the longitudinal cut removes the outer layer. The elemental distribution analysis by µ-EDXRF showed that, in Ariete cultivar, Se is mostly homogeneously distributed in the grain (Figure 2) regardless of treatments with sodium selenate or sodium selenite. Nevertheless, selenite seems to be more effective in Se enrichment which is validated by the concentration observed in brown grain—12.1 and 16.7 mg kg^−1^, for 75 and 100 g Se ha^−1^, respectively.

The Se map of the control grains shows a fairly homogeneous distribution of Se within the grain with somewhat higher values near the bran, albeit with a very low concentration, especially in the Ariete cultivar. In the Ceres cultivar, regarding the treatments with 75 and 100 g Se ha^−1^, the Se distribution seems to favor accumulation in the periphery, perhaps in the bran (Figure 2).

## 3. Discussion

After the second application of Se, net photosynthesis (P_n_) showed to be quite stable in both cultivars, although with significant rises in Ceres at the greatest doses of both Se-forms. On the other hand, g_s_ and E tended to show higher values. This was clearer at the maximal doses of selenite and selenate, particularly in Ariete (Figure 1). Other authors showed that the application of Se in rice fields enhanced photosynthesis by increasing P_n_, E and the intercellular CO_2_ concentration (C_i_). These findings were in line with the rise of the photochemical efficiency of photosystem II (F_v_/F_m_), and a reduced F_o_ [30], thus showing that both leaf gas exchanges and chlorophyll fluorescence analysis can be used to monitoring the photosynthetic functioning of Se-treated rice plants.

At a cellular level, chloroplasts are one of the first target organs of environmental stress [31], leading to changes in photosynthetic rates. However, the addition of adequate levels of Se can mitigate some damage at the photosynthetic apparatus [32], and even counteract for expected global warming that will affect grain production, especially in rice (see [19]). When investigating the effects of Se (0, 1, 5 and 25 μM Na_2_SeO_3_) in photosynthesis, antioxidative capacity and ion homeostasis in maize under salinity, Jiang et al. [32] showed that Se (1 μM) relieved the salt-induced inhibitory effects on the plant growth and development while increasing the net photosynthetic rate and alleviating the damage to chloroplast ultrastructure induced by NaCl. They also showed that P_n_ values decrease as [As] increases, while g_s_ remains stable when 1 and 5 μM Se are used, decreasing 30% with the highest Se application.

As stated above, although the P_n_ values in Ariete showed only a marginal increase in Se-treated plants, Ceres presented a significant (although with a maximal increase of ca. 6%) P_n_ rise, thus indicating a somewhat Se stimulating effect at the maximum levels of both Se-forms, which could be positively reflected in greater yields, in line with the higher number of tillers per plant, more grains per panicle, bigger grains and higher yields found in rice plants sprayed with selenite [33,34]. In fact, there are a wealth of reports pointing to beneficial Se impacts in plants, including increase in photosynthesis [35], enhancement of plant growth and development [36], to increased crop yield and quality [37,38,39].

Notably, in both rice cultivars greater g_s_ and E values were observed in Se-treated plants, particularly at the highest dose of 100 g Se ha^−1^, stronger values were recorded in Ariete. These findings contrast with those observed for maize in which both g_s_ and E values decreased as Se concentration increased [32]. Furthermore, in Ariete the P_n_ and g_s_ variation led to a reduction in iWUE by 41% and 32% compared to control plants, for the 100 g-Se.ha^−1^ selenite and selenate, respectively, whereas in Ceres no significant changes were observed as regards iWUE. Again, our results with rice contrasted with those for *Raphanus sativus*, where all the treatments with Se (sodium selenate and sodium selenite) and two forms of application (soil and foliar application) resulted in increasing stomatal conductance, transpiratory rate, instantaneous efficiency of water use and intrinsic efficiency of water use when compared with control plants [40], thus denoting a species/cultivar dependent responsiveness to Se fertilization. In fact, the link between ecophysiology data and Se accumulation in the rice grain is crucial to understand whether the impacts at leaf gas exchanges (particularly photosynthesis) are reflected in yield and quality (i.e., Se accumulation). Still, the dose of 100 g Se.ha^−1^ can be applied to the Ariete and Ceres cultivars without compromising the photosynthetic machinery, which is quite promising as regards the maximization of Se absorption.

In general, there was a progressive increase in Se in the paddy grains as the foliar concentrations of Se increased (Table 1) and the highest levels were observed in Ceres cultivar with selenite application, although the literature points out a diversity of results mostly depending on the rice cultivars, type of experiment (field research, pot or hydroponic cultures), Se forms and respective concentrations used, type of fertilization (soil or foliar), and soil and climatic variables when experiments are conducted in open fields.

The Se concentrations in rice with foliar spray of selenite or selenate at the late tillering (LT) or full heading (FH) stages show that, in both stages, selenate was much more effective in grain enrichment than selenite [41]. For example, the concentration in rice grains when selenite was applied was 0.44 and 1.29 mg kg^−1^ for LT and FH, respectively, whereas for selenate these values were 0.78 and 2.71 mg kg^−1^, for similar stages. In the same context it was observed that the use of Se under selenite and selenate forms in rice, on both foliar and soil applications, showed interesting and diversified patterns of accumulation [10]. In fact, soil application of selenite pointed out a preferential Se accumulation in roots than in shoots and grain, compared with selenate. Although both forms increased Se grain content, selenate in parallel with soil application are the best choices for maximum enrichment.

Two different cultivars of rice were submitted to foliar fertilization by various concentrations of selenite and selenate (25, 50, 75 and 100 g Se.ha^−1^) at different stages of rice development, i.e., booting, anthesis and milky grain phase. After harvest a 4.9–7.1-fold increase in Se content was observed in grain from plants sprayed with selenate, while a 5.9–8.4-fold increase was noted for selenite [42]. These findings are in agreement with previous reports using four rice genotypes (Ariete, Albatros, OP1105 and OP1109) in field trials with foliar fertilization ranging between 0 and 300 g Se.ha^−1^ for sodium selenite and sodium selenate [14]. At the end of the plant cycle, the most prominent results regarding Se accumulation in the grains were achieved with selenite applications without relevant inhibitory effects on yields [14].

Furthermore, the use of selenite fertilizer via foliar application was more effective in Se rice grain accumulation in four different genotypes than selenate, although the highest concentrations of both fertilizers (120 and 180 g Se.ha^−1^) increased total lipids in all cultivars, mainly oleic, linoleic and palmitic acids [43].

Regarding the effects of selenite and selenate fertilization on the elemental composition of some micro and macronutrients in rice grains, the differences related with the type of application i.e., foliar or via soil, can be stressed once again. For example, in rice, the Cu and Fe levels in the grains are not significantly different (*p* ≤ 0.05) when soil fertilization occurs at a dose of 0.75 mg kg^−1^, and regardless of the use of selenate or selenite. Conversely, when foliar fertilization was used at a concentration of 50 µm L^−1^, the highest levels of these elements were always observed for selenate [10]. The same authors point out that the concentrations of Zn are similar, in general, regardless of the type of fertilization, although the highest value was found via soil fertilization for selenate [10].

Selenium in the form of selenite at a concentration of 150 mg ha^−1^ and 300 mg ha^−1^, was used via foliar spraying on two rice cultivars (MAK and IR-87864) in paddy fields of Mozambique and the analysis of the flour derived from the whole grain revealed a small decrease in Zn flour (9.06 mg kg^−1^ in the case of 150 Se treatment; 9.11 mg kg^−1^ with 300 Se treatment) compared with the control (9.79 mg kg^−1^). Regarding the IR cultivar, the control and plants treated with 150 mg Se ha^-1^ contained 12.6 mg kg^−1^, while plants treated with the double concentration only exhibited 10.6 mg kg^−1^ [18]. Our rice cultivars show a significant decrease in Zn content, when treated with both selenite and selenite, although our analysis was performed at grain level (paddy), thus indicating that if flour was obtained the pattern would be the same. The same authors [18] also point out that remarkable decreases in the Zn content were observed when the flour was obtained from polished grains.

The reduction in values of a* and b* values throughout the grain processing is in agreement with the removal of outer pigment-rich layers, a polished grain with almost starch remaining, although the degree of milling (DOM) and milling time were fundamental in the whole process. Brightness (L* value) of the raw rice kernel and rice flour [44] increased until a DOM of approximately 15% when bran and outer endosperm were removed, which agrees with our trend regarding L* value. They also noted that further milling did not affect rice brightness [44]. Nevertheless, external factors such as post-harvest processes (storage and drying) can influence grain color and should also be considered [45] when performing this type of analysis.

The elemental location within the grain, in particular Se, is important in order to understand its role in the development of the seed [46]. In addition, the knowledge of Se location and speciation is important, because the anti-cancer effects depend on its speciation [47].

Our results show that Se is distributed evenly on the endosperm, although in particular cases a peripheral location was noted, corresponding to the bran (Figure 2). A synchrotron X-ray fluorescence (XRF) analysis of the Se distribution in a mature rice grain slice showed that Se was present throughout the endosperm but was particularly located in the Ovular Vascular Trace (OVT) region (the site of nutrient influx into the grain), the pericarp/aleurone layer, and the embryo [48].

Rice plants growing in a Se-rich environment allowed the demonstration that, within the grain, Se was concentrated in the bran layer, with concentrations almost twice those observed in polished grains. Speciation of the mature grain revealed that Se was present primarily in organic forms, mainly SeMet with lower concentrations of SeMeSeCys and SeCys [49], also emphasizing that that Se concentrations in rice decreased in the order rice straw > bran > whole grain > polished rice > husk

Regarding the polishing of rice grain, it is well known that some vitamins and minerals are lost with the removal of husk and bran. The polishing process similar to that of commercial mills, i.e., 8–10% loss of grain weight, reduced the concentration of Fe, Mg, P, K and Mn by 60–80% in all the studied genotypes, while with a smooth polishing (3–5% weight loss) the losses were reduced to a variable degree, according to the ease of polishing the genotypes. The concentration of other elements, e.g., Zn, S, Ca, Cu, Mo and Cd, showed comparable reductions (< 30%) irrespective of polishing technique or ease of polishing [50]. A much greater proportion of Fe is lost by polishing than is the case for Zn. This reflects their distribution pattern within the grain, Fe being mainly confined to the outer grain layers while Zn is more evenly distributed [50].

These findings agree with our results since the losses of Cu and Zn were the lowest ones with reductions, overall, ranging between 25% and 40%, while Fe, Ca and K showed decreases generally above 70%, although the DOM was ignored in our case. The loss of Se and Mg also occurred, seldom exceeding 50%. In Ariete, the Se losses range between approximately 20% and 30% and in three cases very small enrichments were detected which are probably related to the distribution of the element within the grain. The losses in Ceres grains were more pronounced, mainly in selenate fertilization using the two highest concentrations with percent values around 60%.

Three different pigmented rice genotypes cultivars were used to access the loss of minerals during the polishing process. The results indicate a huge loss regarding the concentrations of the elements (10) studied in the flour vis-a-vis similar concentrations observed in the flour derived from rough grains [51]. For example, Ca levels decrease between 44% and 61% while P levels decrease between 45% and 80%. Regarding micronutrients, Fe decreases between 35% and 70% whereas Zn decreases between 42% and 54% [51].

The effect of milling on the elemental composition of 16 brown rice cultivars was studied [52]. The results indicated that when the DOM was increased twice from 5% to 10%, elemental percent losses also increased significantly in the majority of the cases >50%. Only Mo and Zn exhibited lower losses with percentages ranging between 24% and 34% and 2.4 and 4.6%, respectively [52]. Thus, it seems that the different patterns of polishing losses of minerals reflected their distribution within the grain, indicating that a careful selection of genotypes with reduced polishing losses must be taken into account [50].

All the data discussed above point out the potential health benefits for the consumption of whole brown rice grain, avoiding in this way the huge loss of nutrients, dietary fiber, and bioactive components and contributing to decrease the prevalence of several chronic diet-related diseases such as type 2 diabetes, hypertension, obesity which are increasing worldwide [53]. Furthermore, studies have suggested that brown rice is associated with a wide spectrum of nutrigenomic implications such as having anti-diabetic, anti-cholesterol, cardio protective and antioxidant properties, due to the presence of various phytochemicals that are mainly located in bran [54]. Thus, it will be important in the near future to continue to develop brown rice-based diets and alert consumers about the importance of changes in eating habits.

## 4. Materials and Methods

### 4.1. Experimental Fields

The trial was conducted under field conditions at the experimental station of the Rice Technological Center (COTARROZ), located in Salvaterra de Magos (39°02′21.8″ N; 8°44′22.8″ W). Two rice cultivars, Ariete and Ceres, were tested. The trial duration was from 30 May to 2 November of 2018 and was characterized by maximum and minimum average temperatures of 29 °C and 18 °C (with absolute maximum and minimum values of 47 °C and 9 °C, respectively), with an average rainfall of 0.67 mm and a daily maximum of 17 mm. Biofortification was carried throughout three foliar sprayings with solutions of sodium selenate (Na_2_SeO_4_) and sodium selenite (Na_2_SeO_3_) spread during the life cycle of the plant. Cultivars were sown in six-row plots with specific equipment for sowing trials and then immediately irrigated.

The experimental design was performed in randomized blocks and a factorial arrangement (5 concentrations × 2 forms selenium × 2 cultivars × 4 replicates = 80 plots). The plot size for each replication was 9.6 m^2^ (8 m length × 1.2 m width). The agronomic management of trials, namely the application of nitrogen fertilizers, control of weeds, insect pests and diseases and the water management (irrigation) were those recommended and typically used for the rice crop in this region. Furthermore, the accurate characterization, regarding soil characteristics and irrigation water, of the studied area was identical to that described in a previous and recently published work [42], where some climate parameters were also taken into account.

The agronomic Se biofortification comprised three distinct phases. The first Se application occurred at the end of booting, the second one at anthesis and the third at the milky grain stage. The plants were sprayed with Na_2_SeO_4_ and Na_2_SeO_3_ in different concentrations (25, 50, 75 and 100 g Se ha^−1^). Controls plants were not sprayed at any time with Na_2_SeO_4_ or Na_2_SeO_3_. Grain harvest occurred on 02 November 2018. In both cultivars, the analysis occurred in the paddy, brown, and white rice grains. In polished rice the husk, bran and germ were removed; thus, it was expected that several minerals, biotin, niacin, proteins, among other constituents, were lost during this process. In brown rice only the husk is removed (sometimes partially the bran), while rough rice or paddy rice is the whole rice grain with its hulls— it comes directly from the field after harvest.

### 4.2. Leaf Gas Exchange Measurements

Leaf gas exchange parameters were determined in the trial field, using 4–6 randomized leaves per treatment, on 12 September, following the methods described elsewhere [55]. Leaf rates of net photosynthesis (P_n_), stomatal conductance to water vapor (g_s_) and transpiration (E) were obtained under photosynthetic steady-state conditions after ca. 2 h of illumination (in mid-morning). A portable open-system infrared gas analyzer (Li-Cor 6400, LiCor, Lincoln, NE, USA) was used under environmental conditions, with external CO_2_ (*ca.* 400 mg. Kg^−1^) and photosynthetic photon flux density (PPFD) of ca. 1000 µmol m^−2^ s^−1^. Leaf instantaneous water-use efficiency (iWUE) was calculated as the P_n_-to-E ratio, representing the units of assimilated CO_2_ per unit of water lost through transpiration.

### 4.3. Atomic Absorption Spectrometry

After harvest, 1 g of each sample, previously dried at constant weight, was weighed and placed in a 50-mL Erlenmeyer. An acid digestion procedure was performed with a mixture of HNO_3_−HClO_4_ (4:1) according the methods described elsewhere [56,57]. This process consists of two sequential steps, the first one using 10 mL of HNO_3_, and the second one using 2 mL of HNO_3_ plus 3 mL of HClO_4_. In both cases, samples were digested at 100–150 °C until total evaporation. The residue of the final digestion was diluted in a 2% HCl solution and filtered (Whatman No. 4) into a 50-mL volumetric flask. The standard solution or the blank was prepared with 2% HCl and then analyzed. The concentration of macronutrients (Ca, K and Mg) in the paddy, brown and white grains was measured by a flame process, using an atomic absorption spectrophotometer model Perkin Elmer AAnalyst 200 fitted with a deuterium background corrector, with the AA WinLab software program.

### 4.4. Analysis of Micronutrients and Se Content and Location in the Grain Tissues

The quantification of Cu, Fe, Zn and Se, plus the localization of Se in the grains, harvested from controls and after foliar spraying with Na_2_SeO_4_ (100 g Se ha^−1^) and Na_2_SeO_3_ (100 g Se ha^−1^) was determined by Energy Dispersive X-Ray Fluorescence (µ-EDXRF system, M4 Tornado™, Bruker, Germany), according to Cardoso et al. [58]. The X-ray generator was operated at 50 kV and 100 µA without the use of filters, to enhance the ionization of low-Z elements. For a better quantification of Se and other medium to high atomic weight elements, a set of filters between the X-ray tube and the sample, composed of three foils of Al/Ti/Cu (with a thickness of 100/50/25 µm, respectively), was used. All the measurements with filters were performed with a 600-µA current. Detection of fluorescence radiation was performed by an energy-dispersive silicon drift detector, XFlash™, with 30 mm^2^ sensitive area and energy resolution of 142 eV for Mn Kα. To better measure the distribution mapping of Se, the rice grain was cut, at the equatorial region, into slices with a stainless-steel surgical blade. Measurements were carried out under 20 mbar vacuum conditions and performed directly on the two sides of grains, in the mapping mode. Quantification analysis was performed on the obtained maps by selecting the area corresponding to the entire grain.

The reference standard material used in this work to assess the quality of the quantification procedure was the ERM BB186 (Pig Kidney) with a certified Se concentration of 10.3 ± 0.5 mg Kg^−1^, which is in the same range as the values found in this work. These standards were also used to infer the lower detection and quantification limits for Se which are around 1 and 3 mg kg^−1^, respectively. Regarding Cu, Fe and Zn quantification, a set of standard reference materials were used—Orchard Leaves (NBS 1571) and Poplar Leaves (GBW 07604), with recovery rates ranging between 97% and 99%. The lower detection limit for Fe is 5 mg kg^−1^, while for Cu and Zn it is 2 mg kg^−1^.

### 4.5. Colorimetric Analysis

The color parameters, using fixed wavelength, adopted the methodology described by Ramalho et al. [59]. Brightness/brightness (L) and chromaticity parameters (a* and b* coordinates) were obtained with a Minolta CR 300 colorimeter (Minolta Corp., Ramsey, NJ, USA) coupled to a sample vessel (CR-A504). Using the illuminant D_65_, the system of the *Commission Internationale d’Éclaire* (CIE) was applied. The parameter L represents the brightness of the sample, translating the variation of the tonality between dark and light, with a range between black (0) and white (100). Parameters a* and b*, indicate color variations. The value of a* characterizes coloring in the region from green (−60) to red (+60) and the value b* indicates coloring in the range of between blue (−60) and yellow (+60). The approximation of these coordinates to the null value translates neutral colors like white, gray and black. Measurements were carried out in quadruplicate in the grains of rice at harvest.

### 4.6. Statistical Analysis

Statistical analysis of the data was performed with the IBM SPSS Statistics 20 program, through a one-way analysis of variance and the Tukey’s test for mean comparison. A value of *p* ≤ 0.05 was considered to be significant.

## 5. Conclusions

The foliar fertilization with Se is an effective method to enrich the Se content of the rice grains despite the losses verified for all the studied elements in polished grains vis-a-vis rough grains. The selenium toxicity threshold was not exceeded, as shown by different evaluations regarding the eco-physiological state of the plant evaluated through leaf gas exchanges. The use of Se fertilizers, at the concentrations used, did not seem to influence the concentration of the different elements in the paddy grain, except Zn. Selenite is the best choice regarding the Se enrichment and in polished grains the concentrations of Se submitted to the different treatments are similar, regardless of the cultivars. The location of Se in dehusked grains (brown rice) showed that, in Ariete cultivar, Se is mostly homogeneously distributed, while in the Ceres cultivar the Se distribution seems to favor accumulation in the periphery, perhaps in the bran. The Se map of the control grains show a fairly homogeneous distribution of Se within the grain with somewhat higher values near the bran, albeit with a very low concentration, especially in the Ariete cultivar.

## Figures and Tables

**Figure 1 plants-10-00288-f001:**
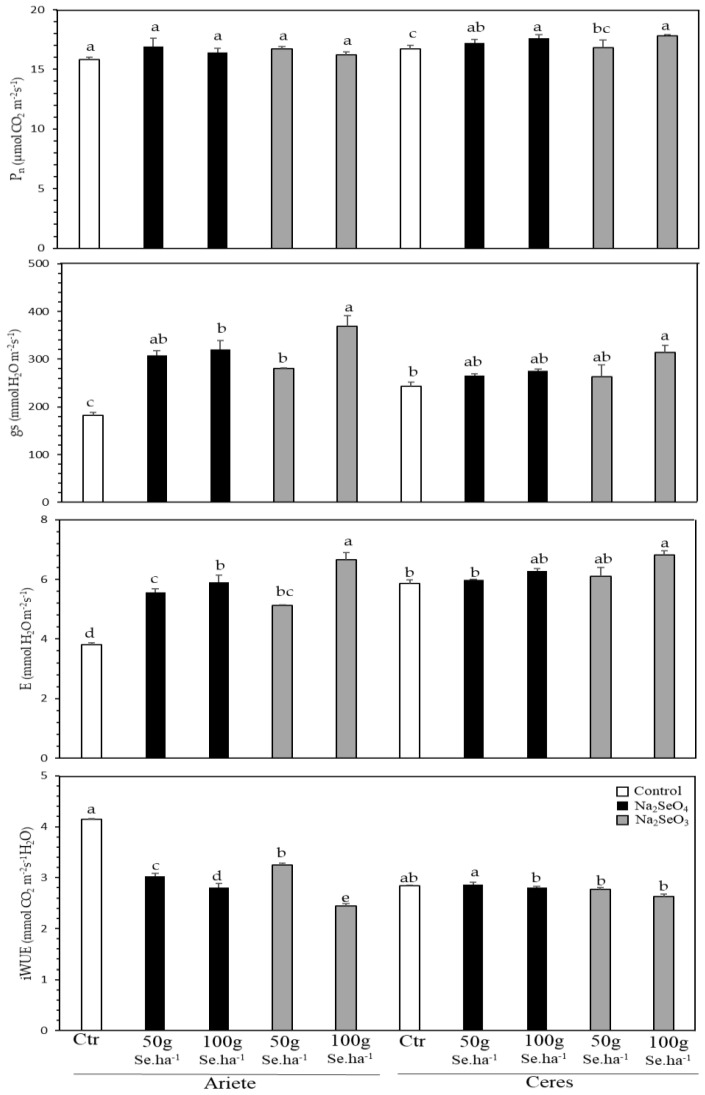
Leaf gas exchange parameters—net photosynthesis (P_n_), stomatal conductance to water vapor (g_s_), transpiration (E) rates, and instantaneous water use efficiency (iWUE) in leaves of *O. sativa*, cultivars Ariete and Ceres. Different letters indicate significant differences between treatments, for each cultivar (*p* ≤ 0.05); average ± standard errors (*n* = 4−6).

**Figure 2 plants-10-00288-f002:**
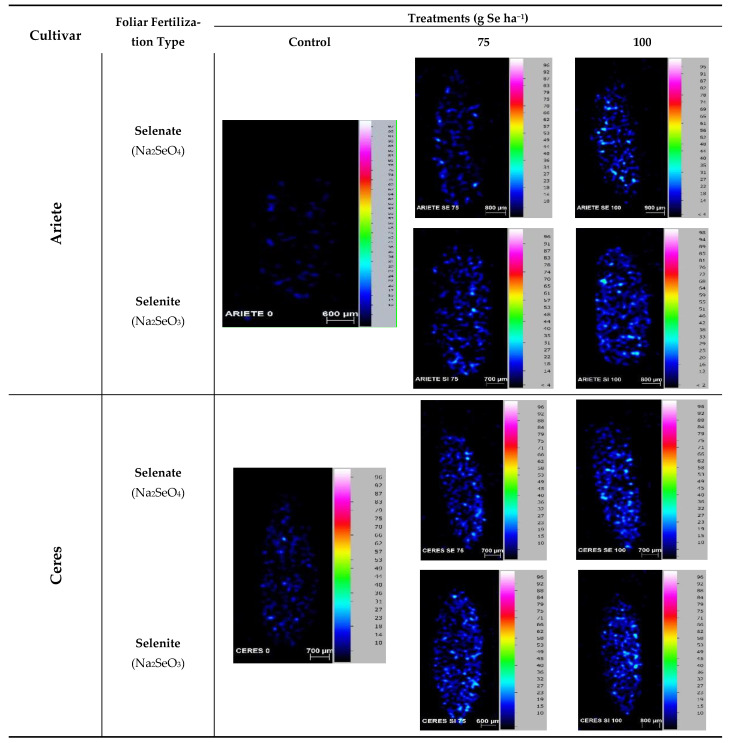
Localization of Se in rice grains of Ariete and Ceres cultivars after foliar spraying with the highest concentrations of sodium selenite and sodium selenate. Control grains did not receive any type of Se enrichment.

**Table 1 plants-10-00288-t001:** Selenium, zinc, iron, copper, calcium, magnesium and potassium concentrations in the paddy grains of *O. sativa* cultivars.

Treatments (g Se ha^−1^)			Se	Zn	Fe	Cu	Ca	Mg	K
Ariete	Na_2_SeO_4_	Control	1.69 ± 0.20 ^c^	47.0 ± 11.4 ^a^	34.9 ± 2.4 ^a^	5.13 ± 0.87 ^a^	43.9 ± 7.07 ^a^	443 ± 3.49 ^a^	6300 ± 1000 ^b^
		25	2.66 ± 0.23 ^c^	36.0 ± 2.76 ^a^	35.1 ± 3.02 ^a^	4.85 ± 0.05 ^a^	42.4 ± 7.89 ^a^	448 ± 2.58 ^a^	6300 ± 200 ^b^
		50	4.06 ± 1.38 ^b^	26.8 ± 4.14 ^b^	38.5 ± 5.49 ^a^	5.35 ± 0.93 ^a^	21.9 ± 5.89 ^a^	445 ± 0.40 ^a^	8900 ± 1600 ^a^
		75	6.06 ± 0.42 ^ab^	30.3 ± 4.44 ^a^	35.0 ± 2.36 ^a^	5.19 ± 0.51 ^a^	27.1 ± 6.22 ^a^	418 ± 6.57 ^b^	8200 ± 2100 ^a^
		100	8.10 ± 2.21 ^a^	30.7 ± 6.34 ^a^	36.8 ± 5.90 ^a^	5.94 ± 1.05 ^a^	30.9 ± 8.13 ^a^	415 ± 9.10 ^b^	8700 ± 1400 ^a^
	Na_2_SeO_3_	Control	1.69 ± 0.20 ^c^	47.0 ± 11.4 ^a^	34.9 ± 2.40 ^a^	5.13 ± 0.87 ^a^	43.9 ± 7.07 ^a^	443 ± 3.49 ^a^	6300 ± 1000 ^b^
		25	5.92 ± 2.38 ^ab^	28.3 ± 2.42 ^b^	31.0 ± 6.52 ^a^	5.34 ± 0.63 ^a^	40.0 ± 5.09 ^a^	440 ± 2.77 ^a^	9400 ± 900 ^a^
		50	7.55 ± 1.79 ^ab^	33.0 ± 6.77 ^a^	39.6 ± 3.12 ^a^	5.30 ± 0.76 ^a^	44.7 ± 8.83 ^a^	449 ± 9.00 ^a^	8700 ± 2300 ^a^
		75	9.88 ± 2.83 ^a^	31.6 ± 3.28 ^a^	35.8 ± 5.71 ^a^	5.79 ± 0.17 ^a^	45.3 ± 6.11 ^a^	441 ± 4.45 ^a^	9000 ± 1400 ^a^
		100	13.0 ± 2.52 ^a^	29.6 ± 3.12 ^b^	39.2 ± 12.7 ^a^	5.02 ± 0.84 ^a^	39.8 ± 8.98 ^a^	456 ± 5.88 ^a^	6900 ± 300 ^b^
Ceres	Na_2_SeO_4_	Control	2.01 ± 0.32 ^c^	69.7 ± 1.74 ^a^	42.7 ± 4.59 ^a^	7.43 ± 0.81 ^a^	76.1 ± 2.99 ^a^	424 ± 11.2 ^ab^	6700 ± 600 ^ab^
		25	4.11 ± 0.76 ^b^	36.5 ± 3.90 ^b^	37.8 ± 8.46 ^a^	7.74 ± 0.98 ^a^	59.1 ± 5.29 ^b^	408 ± 5.90 ^b^	8700 ± 700 ^a^
		50	5.67 ± 1.63 ^a^	33.6 ± 9.27 ^b^	40.1 ± 5.59 ^a^	6.50 ± 2.21 ^a^	51.1 ± 0.81 ^b^	453 ± 20.3 ^a^	6100 ± 700 ^b^
		75	8.66 ± 3.25 ^a^	31.4 ± 1.94 ^b^	43.7 ± 8.92 ^a^	6.53 ± 1.19 ^a^	57.3 ± 2.19 ^b^	418 ± 12.9 ^b^	5500 ± 500 ^b^
		100	11.6 ± 3.94 ^a^	49.3 ± 16.0 ^ab^	50.0 ± 2.05 ^a^	7.10 ± 1.07 ^a^	67.8 ± 2.31 ^ab^	419 ± 8.92 ^b^	7100 ± 500 ^ab^
	Na_2_SeO_3_	Control	2.01 ± 0.32 ^c^	69.7 ± 1.74 ^a^	42.7 ± 4.59 ^a^	7.43 ± 0.81 ^a^	76.1 ± 2.99 ^a^	424 ± 11.2 ^b^	6700 ± 600 ^a^
		25	6.30 ± 1.22 ^b^	39.2 ± 5.22 ^a^	51.0 ± 9.84 ^a^	7.26 ± 1.40 ^a^	40.6 ± 6.58 ^b^	437 ± 8.57 ^b^	8200 ± 900 ^a^
		50	11.8 ± 3.85 ^a^	36.7 ± 7.97 ^a^	40.2 ± 4.28 ^a^	6.11 ± 1.12 ^a^	56.7 ± 5.78 ^b^	457 ± 4.38 ^a^	6300 ± 600 ^a^
		75	10.3 ± 2.01 ^a^	33.4 ± 2.82 ^a^	50.0 ± 8.97 ^a^	5.75 ± 0.71 ^a^	57.8 ± 2.37 ^b^	429 ± 3.88 ^b^	6000 ± 1100 ^a^
		100	12.7 ± 2.91 ^a^	38.0 ± 1.99 ^a^	45.2 ± 8.25 ^a^	5.99 ± 0.43 ^a^	58.4 ± 7.53 ^b^	444 ± 1.54 ^b^	6800 ± 1500 ^a^

Different letters within the same column, cultivar and treatment indicate significant differences (*p* ≤ 0.05). Average values are expressed in mg kg^−1^ ± standard deviation (*n* = 3).

**Table 2 plants-10-00288-t002:** Selenium, zinc, iron, copper, calcium, magnesium and potassium concentrations in the brown grains of *O. sativa* cultivars.

Treatments (g Se ha^−1^)			Se	Zn	Fe	Cu	Ca	Mg	K
Ariete	Na_2_SeO_4_	Control	1.75 ± 0.28 ^c^	29.5 ± 6.08 ^a^	24.3 ± 1.79 ^a^	5.48 ± 0.65 ^a^	25.9 ± 1.74 ^a^	448 ± 18.6 ^a^	5300 ± 100 ^b^
		25	2.86 ± 0.37 ^c^	27.9 ± 3.41 ^a^	19.4 ± 3.50 ^a^	5.34 ± 0.40 ^a^	20.9 ± 3.88 ^a^	438 ± 6.87 ^a^	8500 ± 1600 ^a^
		50	3.74 ± 0.39 ^ab^	28.5 ± 6.31 ^a^	18.5 ± 5.47 ^a^	5.05 ± 1.35 ^a^	11.3 ± 3.62 ^b^	453 ± 4.22 ^a^	7000 ± 2000 ^a^
		75	5.51 ± 1.50 ^ab^	24.9 ± 1.18 ^a^	14.6 ± 2.60 ^a^	4.41 ± 0.57 ^a^	18.4 ± 4.13 ^a^	447 ± 6.74 ^a^	6000 ± 1400 ^a^
		100	7.17 ± 1.84 ^a^	28.9 ± 4.74 ^a^	15.9 ± 2.44 ^a^	4.96 ± 0.12 ^a^	13.1 ± 2.49 ^b^	425 ± 11.8 ^a^	6600 ± 800 ^a^
	Na_2_SeO_3_	Control	1.75 ± 0.28 ^c^	29.5 ± 6.08 ^a^	24.3 ± 1.79 ^a^	5.48 ± 0.65 ^a^	25.9 ± 1.74 ^b^	448 ± 18.6 ^a^	5300 ± 100 ^b^
		25	2.15 ± 0.14 ^c^	27.6 ± 6.06 ^a^	19.7 ± 1.69 ^a^	4.20 ± 0.48 ^a^	26.8 ± 1.34 ^b^	443 ± 5.46 ^a^	5700 ± 1500 ^a^
		50	6.76 ± 1.32 ^b^	28.3 ± 5.73 ^a^	17.5 ± 0.96 ^a^	7.67 ± 5.18 ^a^	45.1 ± 11.0 ^a^	454 ± 7.04 ^a^	4800 ± 700 ^a^
		75	12.1 ± 4.13 ^ab^	27.6 ± 4.21 ^a^	19.8 ± 1.00 ^a^	4.96 ± 0.71 ^a^	33.0 ± 2.78 ^a^	454 ± 2.84 ^a^	7700 ± 600 ^a^
		100	16.7 ± 4.30 ^a^	26.1 ± 6.07 ^a^	14.1 ± 1.15 ^a^	4.92 ± 0.62 ^a^	29.4 ± 5.92 ^ab^	446 ± 2.14 ^a^	7300 ± 2800 ^a^
Ceres	Na_2_SeO_4_	Control	2.11 ± 0.25 ^c^	44.6 ± 7.53 ^a^	22.4 ± 1.56 ^a^	5.42 ± 0.62 ^a^	49.0 ± 7.51 ^a^	447 ± 8.03 ^a^	7800 ± 0300 ^a^
		25	3.79 ± 0.29 ^b^	28.2 ± 5.90 ^b^	16.4 ± 3.06 ^a^	4.96 ± 0.29 ^a^	40.4 ± 4.37 ^a^	442 ± 6.28 ^a^	6300 ± 1200 ^a^
		50	3.78 ± 0.83 ^b^	28.7 ± 4.45 ^b^	18.7 ± 1.96 ^a^	585 ± 1.52 ^a^	38.7 ± 2.80 ^a^	452 ± 5.53 ^a^	7400 ± 1000 ^a^
		75	6.69 ± 2.31 ^ab^	30.8 ± 2.47 ^b^	21.6 ± 1.21 ^a^	5.13 ± 0.66 ^a^	34.6 ± 3.04 ^a^	446 ± 11.5 ^a^	7700 ± 2000 ^a^
		100	9.44 ± 2.67 ^a^	39.0 ± 6.26 ^a^	21.5 ± 1.32 ^a^	5.78 ± 0.75 ^a^	35.6 ± 6.76 ^a^	454 ± 3.17 ^a^	9000 ± 0700 ^a^
	Na_2_SeO_3_	Control	2.11 ± 0.25 ^c^	44.6 ± 7.53 ^a^	22.4 ± 1.56 ^a^	5.42 ± 0.62 ^a^	49.0 ± 7.51 ^a^	447 ± 8.03 ^a^	7800 ± 300 ^ab^
		25	4.00 ± 1.50 ^b^	34.5 ± 6.99 ^a^	19.5 ± 1.45 ^a^	5.77 ± 0.31 ^a^	35.1 ± 0.13 ^a^	459 ± 6.06 ^a^	6600 ± 1100 ^b^
		50	9.01 ± 1.52 ^ab^	37.6 ± 5.05 ^a^	20.4 ± 0.46 ^a^	5.98 ± 0.82 ^a^	29.9 ± 3.23 ^a^	452 ± 8.31 ^a^	9800 ± 500 ^a^
		75	14.2 ± 5.65 ^a^	32.2 ± 2.79 ^a^	17.7 ± 2.41 ^a^	5.70 ± 0.55 ^a^	32.0 ± 3.34 ^a^	454 ± 4.45 ^a^	8300 ± 1600 ^ab^
		100	17.7 ± 0.71 ^a^	28.5 ± 0.92 ^b^	19.2 ± 2.15 ^a^	5.24 ± 0.34 ^a^	35.0 ± 4.02 ^a^	457 ± 5.34 ^a^	7300 ± 200 ^ab^

Different letters within the same column, cultivar and treatment indicate significant differences (*p* ≤ 0.05). Average values are expressed in mg kg^−1^ ± standard deviation (*n* = 3).

**Table 3 plants-10-00288-t003:** Selenium, zinc, iron, copper, calcium, magnesium and potassium concentrations in the white grains of *O. sativa* cultivars.

Treatments (g Se ha^−1^)			Se	Zn	Fe	Cu	Ca	Mg	K
Ariete	Na_2_SeO_4_	Control	2.06 ± 0.33 ^c^ (+ 21.9%)	32.5 ± 8.77 ^a^ (30.8%)	9.01 ± 2.47 ^a^ (74.2%)	3.84 ± 0.49 ^a^ (25.1%)	14.1 ± 1.34 ^a^ (67.9%)	329 ± 27.4 ^a^ (25.7%)	1700 ± 100 ^a^ (73.0%)
		25	3.08 ± 0.73 ^bc^ (+ 15.8%)	24.4 ± 4.01 ^a^ (32.2%)	7.85 ± 2.06 ^a^ (77.6%)	4.19 ± 0.22 ^a^ (13.6%)	23.0 ± 2.64 ^a^ (45.8%)	309 ± 8.43 ^a^ (31.0%)	1900 ± 300 ^a^ (70.3%)
		50	4.90 ± 2.14 ^ab^ (20.7%)	21.6 ± 3.40 ^a^ (19.4%)	8.66 ± 2.07 ^a^ (77.5%)	4.05 ± 0.54 ^a^ (24.3%)	21.3 ± 5.04 ^a^ (32.8%)	238 ± 27.1 ^a^ (46.5%)	1600 ± 300 ^a^ (81.9%)
		75	5.59 ± 0.75 ^ab^ (7.76%)	18.3 ± 1.23 ^a^ (39.6%)	8.63 ± 1.74 ^a^ (75.3%)	4.02 ± 0.08 ^a^ (22.5%)	15.4 ± 1.47 ^a^ (43.2%)	295 ± 26.9 ^a^ (29.4%)	2200 ± 200 ^a^ (73.7%)
		100	8.22 ± 3.07 ^a^ (+ 1.5%)	22.3 ± 5.16 ^a^ (27.4%)	8.58 ± 2.33 ^a^ (76.7%)	4.29 ± 0.16 ^a^ (27.8%)	10.7 ± 0.46 ^a^ (65.4%)	286 ± 32.6 ^b^ (31.1%)	2000 ± 500 ^a^ (76.6%)
	Na_2_SeO_3_	Control	2.06 ± 0.33 ^c^ (+ 21.9%)	32.5 ± 8.77 ^a^ (30.8%)	9.01 ± 2.47 ^a^ (74.2%)	3.84 ± 0.49 ^a^ (25.1%)	14.1 ± 1.34 ^a^ (67.9%)	329 ± 27.4 ^a^ (25.7%)	1700 ± 100 ^a^ (73.0%)
		25	4.07 ± 0.50 ^ab^ (31.2%)	22.4 ± 3.79 ^a^ (13.8%)	11.0 ± 1.63 ^a^ (64.5%)	4.00 ± 0.03 ^a^ (25.1%)	10.5 ± 2.80 ^a^ (73.8%)	254 ± 18.2 ^b^ (42.3%)	1900 ± 600 ^a^ (80.2%)
		50	5.36 ± 2.37 ^ab^ (29.0%)	18.9 ± 1.13 ^a^ (42.7%)	8.71 ± 1.66 ^a^ (78.0%)	4.24 ± 0.29 ^a^ (20.0%)	15.4 ± 2.33 ^a^ (70.9%)	283 ± 13.1 ^ab^ (37.0%)	1900 ± 100 ^a^ (78.5%)
		75	10.4 ± 3.29 ^a^ (+ 5.3%)	20.4 ± 2.28 ^a^ (35.4%)	11.6 ± 4.60 ^a^ (67.6%)	4.60 ± 0.46 ^a^ (20.6%)	9.60 ± 1.78 ^a^ (78.8%)	266 ± 18.6 ^ab^ (39.7%)	1900 ± 200 ^a^ (78.6%)
		100	8.79 ± 1.33 ^a^ (32.4%)	19.4 ± 1.21 ^a^ (34.4%)	8.17 ± 1.02 ^a^ (79.1%)	3.98 ± 0.65 ^a^ (20.7%)	10.3 ± 0.89 ^a^ (74.1%)	258 ± 24.6 ^b^ (43.4%)	1700 ± 400 ^a^ (74.8%)
Ceres	Na_2_SeO_4_	Control	2.34 ± 0.17 ^c^ (+ 16.4%)	45.8 ± 1.63 ^a^ (34.3%)	8.02 ± 1.79 ^a^ (81.2%)	4.51 ± 0.46 ^a^ (39.3%)	18.9 ± 2.85 ^a^ (75.2%)	293 ± 8.86 ^a^ (30.9%)	2100 ± 100 ^a^ (69.0%)
		25	2.62 ± 0.38 ^c^ (36.2%)	29.2 ± 3.51 ^a^ (20.0%)	7.52 ± 1.14 ^a^ (80.1%)	4.48 ± 0.30 ^a^ (42.1%)	13.1 ± 1.31 ^a^ (77.8%)	304 ± 9.59 ^a^ (25.5%)	2100 ± 100 ^a^ (75.9%)
		50	3.72 ± 0.41 ^a^ (34.4%)	25.5 ± 1.76 ^a^ (24.1%)	7.93 ± 1.63 ^a^ (80.2%)	4.48 ± 0.12 ^a^ (31.1%)	17.1 ± 2.71 ^a^ (66.5%)	308 ± 8.49 ^a^ (32.0%)	1800 ± 400 ^a^ (70.2%)
		75	3.76 ± 1.14 ^a^ (56.6%)	23.1 ± 3.46 ^a^ (26.4%)	10.7 ± 2.07 ^a^ (75.5%)	4.01 ± 0.84 ^a^ (38.6%)	15.9 ± 1.68 ^a^ (72.2%)	301 ± 11.7 ^a^ (28.0%)	2000 ± 600 ^a^ (63.8%)
		100	4.30 ± 1.00 ^a^ (62.9%)	29.5 ± 2.71 ^a^ (40.2%)	8.45 ± 1.36 ^a^ (83.1%)	4.19 ± 0.40 ^a^ (41.0%)	15.2 ± 3.02 ^a^ (77.6%)	264 ± 10.1 ^b^ (37.0%)	1900 ± 200 ^a^ (72.7%)
	Na_2_SeO_3_	Control	2.34 ± 0.17 ^c^ (+ 16.4%)	45.8 ± 1.63 ^a^ (34.3%)	8.02 ± 1.79 ^a^ (81.2%)	4.51 ± 0.46 ^a^ (39.3%)	18.9 ± 2.85 ^a^ (75.2%)	293 ± 8.86 ^a^ (30.9%)	2100 ± 1 ^ab^ (69.0%)
		25	3.88 ± 1.13 ^b^ (38.4%)	21.9 ± 2.22 ^a^ (44.1%)	8.31 ± 2.07 ^a^ (83.7%)	4.03 ± 0.10 ^a^ (45.8%)	18.7 ± 2.45 ^a^ (53.9%)	242 ± 11.3 ^b^ (44.6%)	1600 ± 200 ^b^ (81.1%)
		50	6.09 ± 1.55 ^ab^ (48.4%)	21.9 ± 1.36 ^a^ (40.3%)	6.45 ± 0.84 ^a^ (84.0%)	4.13 ± 0.36 ^a^ (32.4%)	15.4 ± 2.43 ^a^ (72.8%)	267 ± 8.94 ^a^ (41.6%)	1800 ± 400 ^b^ (71.5%)
		75	9.47 ± 2.68 ^a^ (8.05%)	24.2 ± 3.12 ^a^ (27.5%)	5.93 ± 1.12 ^a^ (88.1%)	4.75 ± 0.76 ^a^ (17.4%)	23.9 ± 2.66 ^a^ (58.6%)	276 ± 10.3 ^a^ (35.7%)	2800 ± 300 ^a^ (54.0%)
		100	11.0 ± 1.51 ^a^ (13.4%)	26.2 ± 5.85 ^a^ (31.0%)	6.15 ± 2.43 ^a^ (86.4%)	4.62 ± 0.37 ^a^ (22.9%)	11.8 ± 0.14 ^a^ (79.8%)	224 ± 12.2 ^c^ (49.5%)	1700 ± 100 ^b^ (75.0%)

Different letters within the same column, cultivar and treatment indicate significant differences (*p* ≤ 0.05). Average values are expressed in mg kg^−1^ ± standard deviation (*n* = 3); data between brackets indicate the percent loss by comparison with rough grains. In some few cases an enrichment was observed, which is described by the signal +.

**Table 4 plants-10-00288-t004:** Color parameters of the paddy, brown and white rice grains of *O. sativa*, cultivars.

	Treatments (g Se ha^−1^)	Paddy Rice			Brown Rice			White Rice		
		L	a*	b*	L	a*	b*	L	a*	b*
	Na_2_SeO_4_									
**Ariete**	Control	59.8 ± 2.23 ^a^	5.57 ± 1.44 ^a^	27.7 ± 3.35 ^a^	63.5 ± 2.35 ^a^	3.48 ± 0.44 ^a^	23.1 ± 0.18 ^a^	73.6 ± 1.11 ^a^	−0.80 ± 0.24 ^a^	9.52 ± 0.54 ^a^
	25	58.6 ± 0.81 ^a^	5.97 ± 1.20 ^a^	27.9 ± 3.19 ^a^	63.4 ± 1.52 ^a^	2.81 ± 0.22 ^a^	22.1 ± 0.95 ^a^	74.2 ± 0.32 ^a^	−0.86 ± 0.13 ^a^	9.25 ± 0.70 ^a^
	50	59.3 ± 2.14 ^a^	5.51 ± 1.52 ^a^	26.6 ± 3.31 ^a^	62.6 ± 3.05 ^a^	3.50 ± 0.96 ^a^	23.6 ± 1.51 ^a^	73.5 ± 1.40 ^a^	−0.78 ± 0.12 ^a^	9.73 ± 1.07 ^a^
	75	58.9 ± 1.81 ^a^	5.78 ± 1.56 ^a^	27.6 ± 3.97 ^a^	62.5 ± 2.75 ^a^	3.20 ± 0.43 ^a^	22.8 ± 0.60 ^a^	74.0 ± 1.52 ^a^	−0.80 ± 0.18 ^a^	9.51 ± 0.38 ^a^
	100	59.6 ± 3.33 ^a^	6.07 ± 0.98 ^a^	28.1 ± 2.48 ^a^	63.0 ± 2.00 ^a^	3.16 ± 0.48 ^a^	23.3 ± 0.58 ^a^	72.1 ± 1.66 ^a^	−0.66 ± 0.10 ^a^	10.1 ± 1.01 ^a^
	**Na_2_SeO_3_**									
	Control	59.0 ± 2.17 ^a^	5.78 ± 1.35 ^a^	27.3 ± 2.87 ^a^	63.6 ± 2.17 ^a^	3.45 ± 0.35 ^a^	23.5 ± 0.91 ^a^	74.3 ± 1.12 ^a^	−0.88 ± 0.15 ^a^	8.78 ± 0.60 ^a^
	25	59.9 ± 2.15 ^a^	5.31 ± 1.71 ^a^	27.4 ± 3.64 ^a^	62.3 ± 1.08 ^a^	3.63 ± 1.12 ^a^	23.1 ± 0.87 ^a^	73.9 ± 2.35 ^a^	−0.60 ± 0.20 ^a^	9.24 ± 0.65 ^a^
	50	58.0 ± 2.98 ^a^	5.73 ± 1.89 ^a^	27.0 ± 3.10 ^a^	63.4 ± 2.22 ^a^	3.10 ± 0.43 ^a^	23.8 ± 1.04 ^a^	74.5 ± 0.24 ^a^	−0.82 ± 0.11 ^a^	9.08 ± 0.48 ^a^
	75	59.1 ± 2.10 ^a^	5.76 ± 1.53 ^a^	27.5 ± 2.86 ^a^	64.2 ± 2.79 ^a^	3.13 ± 0.78 ^a^	23.5 ± 0.75 ^a^	74.4 ± 1.23 ^a^	−0.88 ± 0.11 ^a^	9.12 ± 0.34 ^a^
	100	58.4 ± 2.38 ^a^	5.73 ± 1.90 ^a^	27.2 ± 3.90 ^a^	64.0 ± 1.80 ^a^	3.27 ± 0.56 ^a^	23.8 ± 0.43 ^a^	73.8 ± 1.92 ^a^	−0.88 ± 0.04 ^a^	9.22 ± 0.40 ^a^
	**Na_2_SeO_4_**									
**Ceres**	Control	58.6 ± 1.39 ^a^	6.56 ± 0.32 ^a^	32.8 ± 0.88 ^ab^	62.0 ± 2.03 ^a^	3.57 ± 0.83 ^a^	22.8 ± 0.59 ^a^	72.1 ± 1.51 ^a^	−1.02 ± 0.06 ^a^	8.42 ± 0.47 ^a^
	25	57.5 ± 0.88 ^a^	6.98 ± 0.76 ^a^	33.2 ± 1.10 ^a^	63.1 ± 1.82 ^a^	2.85 ± 1.03 ^a^	22.1 ± 0.65 ^a^	74.0 ± 1.19 ^a^	−1.12 ± 0.11 ^a^	7.59 ± 0.86 ^a^
	50	58.3 ± 0.68 ^a^	6.82 ± 0.14 ^a^	33.4 ± 0.42 ^a^	61.7 ± 2.32 ^a^	2.91 ± 0.47 ^a^	23.0 ± 0.50 ^a^	72.4 ± 0.99 ^a^	−0.94 ± 0.20 ^a^	7.61 ± 1.15 ^a^
	75	58.7 ± 0.62 ^a^	6.60 ± 0.27 ^a^	32.0 ± 0.52 ^ab^	62.9 ± 1.76 ^a^	3.03 ± 0.12 ^a^	22.6 ± 0.49 ^a^	74.3 ± 0.60 ^a^	−1.01 ± 0.15 ^a^	7.92 ± 1.02 ^a^
	100	56.5 ± 1.20 ^a^	6.36 ± 0.28 ^a^	31.2 ± 0.43^b^	62.4 ± 2.88 ^a^	3.26 ± 0.57 ^a^	22.7 ± 0.44 ^a^	73.0 ± 1.66 ^a^	−1.03 ± 0.08 ^a^	8.18 ± 0.50 ^a^
	**Na_2_SeO_3_**									
	Control	57.5 ± 1.28 ^a^	6.09 ± 0.62 ^a^	30.4 ± 1.00 ^a^	61.8 ± 0.86 ^a^	3.50 ± 0.44 ^a^	23.1 ± 1.53 ^a^	72.0 ± 1.42 ^a^	−1.03 ± 0.08 ^a^	8.17 ± 1.09 ^a^
	25	57.5 ± 1.12 ^a^	6.36 ± 0.27 ^a^	31.8 ± 0.83 ^a^	63.8 ± 2.85 ^a^	2.93 ± 0.28 ^a^	22.7 ± 0.77 ^a^	72.5 ± 1.67 ^a^	−0.81 ± 0.38 ^a^	8.13 ± 1.58 ^a^
	50	57.8 ± 0.40 ^a^	6.29 ± 0.34 ^a^	31.7 ± 0.46 ^a^	62.9 ± 0.97 ^a^	3.20 ± 1.01 ^a^	23.6 ± 1.35 ^a^	74.7 ± 0.75 ^a^	−0.98 ± 0.11 ^a^	8.26 ± 1.30 ^a^
	75	57.3 ± 1.47 ^a^	6.98 ± 0.14 ^a^	31.5 ± 0.70 ^a^	63.0 ± 1.81 ^a^	3.18 ± 0.85 ^a^	22.8 ± 2.06 ^a^	73.0 ± 1.40 ^a^	−1.07 ± 0.08 ^a^	7.96 ± 0.83 ^a^
	100	56.9 ± 0.78 ^a^	6.51 ± 0.35 ^a^	31.4 ± 1.07 ^a^	63.1 ± 0.38 ^a^	2.90 ± 0.83 ^a^	22.4 ± 0.50 ^a^	74.2 ± 0.47 ^a^	−1.01 ± 0.14 ^a^	8.31 ± 0.60 ^a^

Different letters indicate significant differences between treatments (Na_2_SeO_4_ or Na_2_SeO_3_), for each cultivar (single factor ANOVA test, *p* ≤ 0.05). Average values ± standard error (*n* = 4). Color parameters: L*—lightness; a*—red—green transitions; b*—yellow—blue transitions.

## Data Availability

Not applicable.

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
