# Peer review of "Effect of Rice Grain (Oryza sativa L.) Enrichment with Selenium on Foliar Leaf Gas Exchanges and Accumulation of Nutrients"

_plants, 2021, doi:10.3390/plants10020288_

Round 1
Reviewer 1 Report
comments:
- There is no introduction about why choosing these two rice varieties? What are the differences between?.
- The author suggested dose of 100 g Se.ha-1 can be applied to the Ariete and Ceres cultivar in order to maximize selenium absorption, what happen to grain yield, there is no grain yield data at all in this manuscript since grain yield is one of the most important agronomic traits.
- There are too many tables rather than Figures, at least one Table should be changed into figure..
Reviewer 2 Report
Dear Author, the manuscripts appears fitting with the topics of the journal. Many interesting results have been obtained. Despite that, the topic is not so innovative, and there are other papers in literature about the same topic. In my opinion, the main merit of this paper is that many different treatment (rice cultivar, Se concentration, Se form, processing phase) have been tested. Thus, it is necessary to stress out the differences among the different factor and the interaction between them. Then, discuss the results focusing the attention on the effects of the interactions. Besides, the other main problem of the paper is that the results are not clearly showed. In text and in tables it is not clear which are the factors used for the statistical analysis and the results of it. Before to make tables, it is necessary to decide which factors and which interactions are worth to be analyzed, and then make it clear in the tables. In addition, it would be useful add the result of the statistical analysis (ns, *, **, ***) in the table.
Some other suggestions:
- Introduction: add more information about the biofortification with Se by foliar application.
- Line 92: “Monitoring”
- Line 96: according to “Table 1”, the differences are not statistically significant, thus it is not possible to said that an increment has been registered.
- Line 99-100: this sentence is not clear.
- Line 101-103: According to Table 1, there was a significant increment of Pn in Na2SeO3 (100), Na2SeO4 (50) and (100).
- Line 102:to report the effect of the Se form on the parameter, it is necessary to perform the statistical analysis using also the Se form as a separate parameter.
- Line 104-106: the only significant difference of Gs was detected in Na2SeO3 (100).
- Line 106-108: describe the results obtained and move this sentence to the “discussions”.
- Line 115-116: statistically this happens also in treatments with selenate.
- Line 118-120: Have you performed the statistical analysis using Se form as a factor?
- Line 122-137: it is not possible to compare the results obtained at different processing phases if the statistical analysis was not performed using the processing phase as a factor of variability.
- Line 149-151 and 160-162…: have you used “variety” as a factor of variability in the statistical analysis? If not, it is not possible describe the results in this way.
- Line 166-167: express all the results with the same unit.
- Lin 210-242: The statistical analysis using the processing phase as a factor of variability needs to be performed to compare the results obtained in white and paddy rice grains.
- Line 243-268: have you used variety and processing phase as factors of variability for the statistical analysis? It is not clear.
- Line 285-322: discuss which could be the causes and the effects on plant growth of the differences in gas exchange parameters induced by Se treatment.
- Line 361: correct the unit of ”50 xm L-1”
- Line 463-470: add at least the climatic parameters.
- Results: it would be interesting adding some results about the biomass production.
In conclusion, to make the paper innovative, the “results” and “discussion” sections need to be rewritten trying to emphasize the interaction between the factors of variability.
Kind regards
Reviewer 3 Report
Some comments for the manuscript
Topic of this article is interesting and actual from the nutrition point of view, rice is important food raw material and could be a source of Se in the human diet.
The presentation is clear and the data are obtained by established methods and the results are supported by appropriate and sufficient references.
There are few minor changes needed in the text (See below with line numbers)
L92: Monitoring (monitoring)
L110: 2.2.1. Accumulation of selenium in grains (delete this the subtitle)
L122: dehusked
L161: P < 0.05
L190: 2.3.3. White rice
L200: L224 and 251, draw the lines in the tables correctly
L210, 2.3.4. Comparison between elemental levels in paddy and white rice grains
L226: include one space
L242; 21.9%
Table 4: 2nd row and 1st column with numbers (+21.9%) and 34th row and 5th column with numbers (53.9%)
261; Arite variety
361; 50 m L-1 write the unit correctly
L554: indicate the statistical program with which the data were processed
References: write the years in bold and add the missing numbers doi
L690, doi:
Round 2
Reviewer 2 Report
The authors answered to all my comments and reccomendations. Thus, in my opinion, the manuscript coul be pubblished in the present form.
Author Response
The authors thanks the nice sentences of the reviewer.